# A tightly clustered hepatitis E virus genotype 1a is associated with endemic and outbreak infections in Bangladesh

Trang Nguyen Hoa[1], Saif Ullah Munshi[2], Khanh Nguyen Ngoc[1], Chau Le Ngoc[1], Thanh Tran Thi Thanh[1], Tahmina Akther[2], Shahina Tabassum[2], Nilufa Parvin[3], Stephen Baker[4], Motiur Rahman[1,5]*

1 Oxford University Clinical Research Unit, Wellcome Asia Programme, The Hospital for Tropical Diseases, Ho Chi Minh City, Vietnam, 2 Bangabandhu Sheikh Mujib Medical University, Shahbag, Dhaka, Bangladesh, 3 Sir Salimullah Medical College and Hospital (SSMCH), Dhaka, Bangladesh, 4 Cambridge Institute of Therapeutic Immunology & Infectious Disease, Cambridge University, Cambridge, England, 5 Centre for Tropical Medicine and Global Health, Nuffield Department of Medicine, Oxford University, Oxford, United Kingdom

* mrahman@oucru.org

**Data Availability Statement:** All sequences were submitted to GenBank. Accession no; (MK005535 to MK005551) for the ORF2-3 region, and (MH991993 to MH992013) for the whole genome sequences.

## Abstract

### Background

Hepatitis E virus (HEV) infection is endemic in Bangladesh and there are occasional outbreaks. The molecular characteristics and pathogenesis of endemic and outbreak HEV strains are poorly understood. We compared the genetic relatedness and virulence associated mutations of endemic HEV strains with outbreak strains.

### Methods

We analyzed systematically collected serum samples from HEV immunoglobulin M (IgM) positive patients attended at Bangabandhu Sheikh Mujib Medical University, Dhaka from August 2013 to June 2015. HEV RNA positive samples were subjected to whole genome sequencing. Genotype and subtype of the strains were determined by phylogenetic analysis. Virulence associated mutations e.g. acute viral hepatitis (AVH), fulminant hepatic failure (FHF), chronic hepatitis, ribavirin treatment failure (RTF), B and T cell neutralization epitopes were determined.

### Results

92 HEV immunoglobulin M (IgM) antibody positive plasma samples (43 in 2013–2014 and 49 in 2014–2015) were studied. 77.1% (70/92) of the samples were HEV RNA positive. A 279 bp open reading frame (ORF) 2 and ORF 3 sequence was obtained from 54.2% (38/70) of the strains. Of these 38 strains, whole genome sequence (WGS) was obtained from 21 strains. In phylogenetic analysis of 38 (279 bp) sequence all HEV sequences belonged to genotype 1 and subtype 1a. Further phylogenetic analysis of 21 HEV WGS, Bangladeshi HEV sequences clustered with genotype 1a sequences from neighboring countries. Within genotype 1a cluster, Bangladesh HEV strains formed a separate cluster with the 2010 HEV

**Funding:** This study was supported by the Oxford University Clinical Research Unit, Vietnam.

**Competing interests:** The authors have declared that no competing interests exist.

outbreak strains from northern Bangladesh. 80.9 to 100% of the strains had A317T, T735I, L1120I, L1110F, P259S, V1479I, G1634K mutations associates AVH, FHF and RTF. Mutations in T cell recognition epitope T3, T5, T7 was observed in 76.1%, 100% and 100% of the strains respectively.

## Conclusion

Strains of HEV genotype 1a are dominant in Bangladesh and are associated with endemic and outbreak of HEV infection. HEV isolates in Bangladesh have high prevalence of virulence associated mutations and mutation which alters antigenicity to B and T cell epitopes.

## Introduction

Hepatitis E virus (HEV) infection account for 20 million infections globally per year, 3.3 million of these are symptomatic, and associated with 57,000 deaths [1,2]. The principal burden of HEV disease is in low-middle income countries (LMICs); an estimated 2 billion people live in regions where HEV is endemic [3]. The clinical outcome of HEV infection is varied and largely dependent the genotype of the infecting virus and host factors. The overwhelming majority of HEV infections are self-limiting, but infections in pregnant women are associated with high mortality [4].

HEV is a non-enveloped, single-stranded, positive-sense RNA virus, and a member of the family *Hepeviridae* [5]. The HEV genome is 7.2 kb and is comprised of a short 5' untranslated region (UTR), three open reading frames (ORF1, ORF2, and ORF3), and a 3' UTR, followed by a poly A tail [6]. ORF1, ORF2, and ORF3 encode the nonstructural protein, the viral capsid protein, and a phosphorylated protein, respectively [7,8]. HEV from human, animals, and primates are currently divided into eight genotypes, which includes four genotypes known to cause disease in humans (genotype 1, 2, 3 and 4). These four genotypes are further subdivided into 24 subtypes (genotype 1 (n = 5), 2 (n = 2), 3 (n = 10), and 4 (n = 7)) and a single serotype [5,9].

The various HEV genotypes have a distinct geographic distribution, clinical presentation, and mode of transmission. Genotypes 1 and 2 are found only in humans and are responsible for most HEV cases in Asian LMICs (Myanmar, Pakistan, Bangladesh, India, and Nepal). Genotype 2 is more commonly identified in Latin America and Africa [10]. These genotypes are transmitted fecal-orally, are associated with contaminated water, and typically cause acute infection. Alternatively, genotypes 3 and 4 circulate in various animals (pigs, wild boar, and deer) and are the main cause of HEV infection in high-income countries [10–12]. Genotype 3 is chiefly identified in Europe; genotype 4 is mainly found in more affluent Asian countries including China, Taiwan, Vietnam, and Japan. These genotypes are associated with infection via cross-species transmission often through the consumption of contaminated food [12–14].

The pathogenesis of HEV is poorly understood, although a number of genetic loci have been found to be associated with disease progression. There are several mutations in the HEV genome that have been found to be associated with fulminant hepatic failure (FHF), acute viral hepatitis (AVH), and chronic hepatitis. Additionally, further mutations correlating with ribavirin treatment failure, and B and T cell epitope neutralization have been reported [15–17]. It has been further suggested that these B and T cell epitope mutations are deleterious and decrease the antigenicity of HEV; the mutated epitopes result in variants that evade host B and T recognition [8,18,19].

Bangladesh has a high HEV burden, where small outbreaks are common. The prevalence of HEV antibody among the general population is 22%-60% [20–22]; the incidence of HEV sero-conversion is 40 per 1,000 people per year in rural settings [21]. Drinking municipal tap water, outdoor employment, recent urban travel, and recent contact with a patient with jaundice have been identified as HEV infection risk factors [21,23]. In 2008–2009 a HEV outbreak associated with 4,521 infections and 17 deaths was reported near the capital Dhaka [23]. In 2010 a similar outbreak with 200 hospitalization has been reported in northeast city Rajshahi [22]. Despite the significance of HEV in Bangladesh there has been limited characterization of the HEV circulating in this highly endemic country. Here, we conducted a longitudinal study to identify patients infected with HEV in Bangladesh to characterize the infecting viruses and assess the composition of mutations that may associate with a more severe disease progression.

## Materials and methods

### Study design and population

A prospective cross-sectional study was conducted from August 2013 to June 2015 at Banga-bandhu Sheikh Mujib Medical University (BSMMU) in Dhaka. Patients attending at the virology department of BSMMU with a HEV IgM test request and tested positive were eligible for enrollment. Systematically selected (first four HEV IgM ELISA (Beijing Wantai Biological Pharmacy Enterprise Co., Ltd., Beijing, China) positive patients each month) leftover samples were stored at -86˚C. Plasma samples were unlinked with patient data be replacing the patient ID with a study ID. Samples were shipped to Oxford University Clinical Research Unit (OUCRU), Vietnam for further analysis. The study was approved by the Bangabandhu Sheikh Mujib Medical University (BSMMU) ethical review committee (Approval No. BSMMU/2013/3027).

### Amplification and sequencing of HEV

HEV IgM positive samples were screened for HEV RNA using quantitative PCR (qPCR) [24]. Total viral RNA was extracted from 140ul of plasma using a MagNa pure viral RNA extraction kit (Roche Diagnostics, Basel). RNA was reverse transcribed into cDNA using superscript III reverse transcriptase. cDNA from the qPCR positive samples were used for amplification of the HEV genome by nested PCR. Degenerate oligonucleotide primers were designed to amplify 450–600 bp overlapping fragments to cover the entire HEV genome (S1 File). Additionally, where necessary the 5' and 3' cDNA ends were amplified using RACE PCR as previously described [25]. When required a primer walking strategy was implemented to close gaps in the genome sequences. PCR amplicons were visualized by agarose gel electrophoresis and purified using the QIAamp PCR purification kit (QIAgen, Germany). Purified amplicons were sequenced in both directions using an Applied Biosystem 3130XL Genetic Analyzer. The entire genome sequences (amplified by primer set 1 to 20 (S1 File)) and a 279bp region (position 5972–6319 of GenBank accession no M73218) of ORF2 and ORF3 (part of the '440 bp fragment amplified by primer set 20 (S1 File)) were used for final analysis [26].

### Genome assembly and phylogenetics

To construct the HEV whole genome sequences the overlapping sequences from the PCR amplicons were aligned and a consensus sequence was generated. Additionally, 109 HEV genome sequences representing all genotypes and subtypes including 20 HEV genotype 1 sequences from Asia were accessed from GenBank (S2 File). Sequences generated during this study and the reference sequences were subjected to phylogenetic analysis using Geneious

8.0.5 software [27]. The entire genome sequences and a 279bp fragment of ORF2 and ORF3 were aligned using MUSCLE alignment program within Geneious [28]. Sequence alignments were subjected to Jmodel test to identify the best model for phylogenetic analysis [29]. The suggested nucleotide substitution model (GTR+G+I) was used for phylogenetic analysis in RAxML v7.2.8 (available in Geneious package). To confirm the reliability of phylogenetic tree, bootstrap resampling and reconstruction were repeated on 500 occasions.

Phylogenetic analysis of the whole genome sequences was performed in a two-step process. Firstly, 21 whole genome sequences from the present study and 109 existing genome sequences (including two outbreak sequences from Bangladesh) were analyzed. Secondly, 20 genotype 1a genome sequences from Asia (Bangladesh, India, Nepal, Pakistan, and Myanmar) and 21 genotype 1a HEV genome sequences from the present study were analyzed. All sequences were submitted to GenBank. Accession no; (MK005535 to MK005551) for the ORF2-3 region, and (MH991993 to MH992013) for the whole genome sequences.

### Virulence associated mutation analysis

Known mutations associated with i) FHF, ii) AVR, iii) chronic hepatitis, iv) ribavirin treatment failure, and v) B and T (MHC I and II) cell epitope neutralization were identified in 21 HEV WGS. These mutations were: F179S, A317T, T735I, L1110F, V1120I, F1439Y, C1483W, and N1530T in ORF1 and P259S in ORF2 (FHF and AVH); v1213A in ORF1 (chronic hepatitis); Y1320H, G1634R/K, K1383N, D1384G, K1398R, V1479I, and Y1587F in ORF1 (ribavirin treatment failure); and A317T, T735I, L1110F, V1120I, and G1634K in ORF1 and L477T and L613 in ORF2 (B cell and T cell neutralization). The potential effect of T (MHCI and MHC II) cell and B cell epitope mutations on antigenicity were inferred with published amino acid sequences of wild-type T cell and B cell epitopes [15]; this included epitope T2-T5 in MHCI, T7-T12 in MHCII in T cell epitope, and epitope B2-B8 in B cell epitope.

### Statistical analysis

All data collected in this study were cleaned and entered into an SPSS database. Analysis was performed using Statistical Package for Social Science (SPSS) software (IBM SPSS Statistics 23, NY USA). Categorical variables were compared with Fisher's exact test and continuous variables by Student's t-test and Mann-Whitney U-test as appropriate. A $p$-value <0.05 was considered to be statistically significant.

## Results

### Study population

From August 2013 to June 2015, 92 HEV IgM antibody positive plasma samples were collected (44 (47.8%) in 2013–2014 and 48 (52.2%) in 2014–2015). 53 (57.6%) samples were from female patients, including 24 (45.3%) from pregnant women. The median age of the patients was 25 (IQR; 21–30 years). Of these 92 patients 70 (76.1%) patients were HEV RNA positive (33 men and 37 women (14 pregnant and 23 non-pregnant women)). 84.6% (33/39) of the samples from men were HEV RNA positive compared to 69.8% (37/53) of samples from women ($p$ = 0.08; chi squared test). Among the women patients, 79.3% (23/29) non-pregnant women and 58.3% (14/24) pregnant women were HEV RNA positive. There was no difference in age, sex, HEV RNA positivity among patients recruited during 2013–2014 and 2014–2015 (S3 File).

## Sequencing of HEV

HEV RNA positive plasma samples were subjected to sequencing. A schematic diagram of the DNA amplicons sequenced for each sample is shown in S4 File. At least one DNA fragment could be sequenced in all of the plasma samples. Ultimately, a 279bp fragment of ORF 2–3 (used for genotyping) or an entire genome sequence (when generated) were subjected to phylogenetic analysis. The final data set included 38 (17 from incomplete genome sequence and 21 from complete genome sequence) ORF2-3 sequences and 21 whole genome sequences (1 from pregnant women, 6 from non-pregnant women, and 14 from men).

## Genome assembly and phylogenetics

The average length of novel HEV genomes was 7,215bp (range; 7,202–7,221bp) excluding the 3' poly (A) tail. Each genome had a 17-28bp untranslated region at 5' end, a 66bp untranslated region at 3' end, and a poly A tail of variable length (8-33bp). The complete HEV genome contains 3 open reading frames (ORF); ORF1 was from 28 to 5,109 bp; ORF2 was from 5,147bp–7,129bp and; ORF3 was from 5,106bp– 5,477bp. ORF2 starts after 41 bases of ORF1. ORF3 overlapped ORF1 by four bases and ORF2 by 333bp.

The predicted translated protein of ORF1 was 1,694 amino acids. The NTP-binding domains GVPGSGKS and DEAP in the putative helicase region and the GDD site was found in RDRP regions of all isolates. The ORF1 sequences were highly conserved, with 98.1% identity (min-max; 97.05–97.91%) at the DNA level and 99.5% (min-max; 98.75–100%) identity at the amino acid level. The predicted translated product of ORF2 was 660 amino acids. Identity within ORF2 was 98.6% (min-max; 98.44–97.81%) at a nucleotide level, and 99.5% (min-max, 98.34–100%) at the amino acid level. The translated product of ORF3 product was 124 amino acids. The nucleotide similarity of ORF3 was 99.38% (min-max 98.39–99.19%) and 99.23% (min-max, 97.56–100%) identical at amino acid level.

In a phylogenetic analysis of the 279 bp fragment of ORF2-3 of all 38 Bangladeshi HEV sequences (17 from incomplete genome sequence and 21 from complete genome sequence) from this study clustered in proximity to HEV genotype 1a sequences (S5 File). In a further phylogenetic analysis, the 21 whole genome sequences from this study clustered alongside genotype 1a HEV from India, Burma, and Nepal (Fig 1) and two near complete WGS from a 2010 outbreak in Rajshahi. Notably, the genotype 1 HEV sequences from China, Pakistan, and various African countries formed an independent cluster.

A subsequent phylogenetic analysis with whole genome sequences (n = 20) from counties neighboring Bangladesh found that genotype 1a sequences fell into two main clusters. The first cluster contained HEV from India, Nepal, and Myanmar, the second cluster included the 21 HEV from the present study and two HEV sequence from a previous outbreak in northern Bangladesh (Fig 2); these HEV DNA sequences were >99% identical across the genome.

## Virulence associated mutation analysis

Mutations associated with severe disease in 21 HEV WGS from this study are shown in Table 1. All HEV sequences harbored the A317T, T735I, and L1120I mutations in ORF1; 80.9% had an L1110F mutation in ORF1 and a P259S mutation in ORF2; these are known to be associated with FHF and AVH. None of sequences had a V1213A mutation associated with chronic infection. The ribavirin resistance-associated mutations V1479I and G1634K were also detected in all sequences. Conversely, none of the sequences had the Y1320H, K1383N,

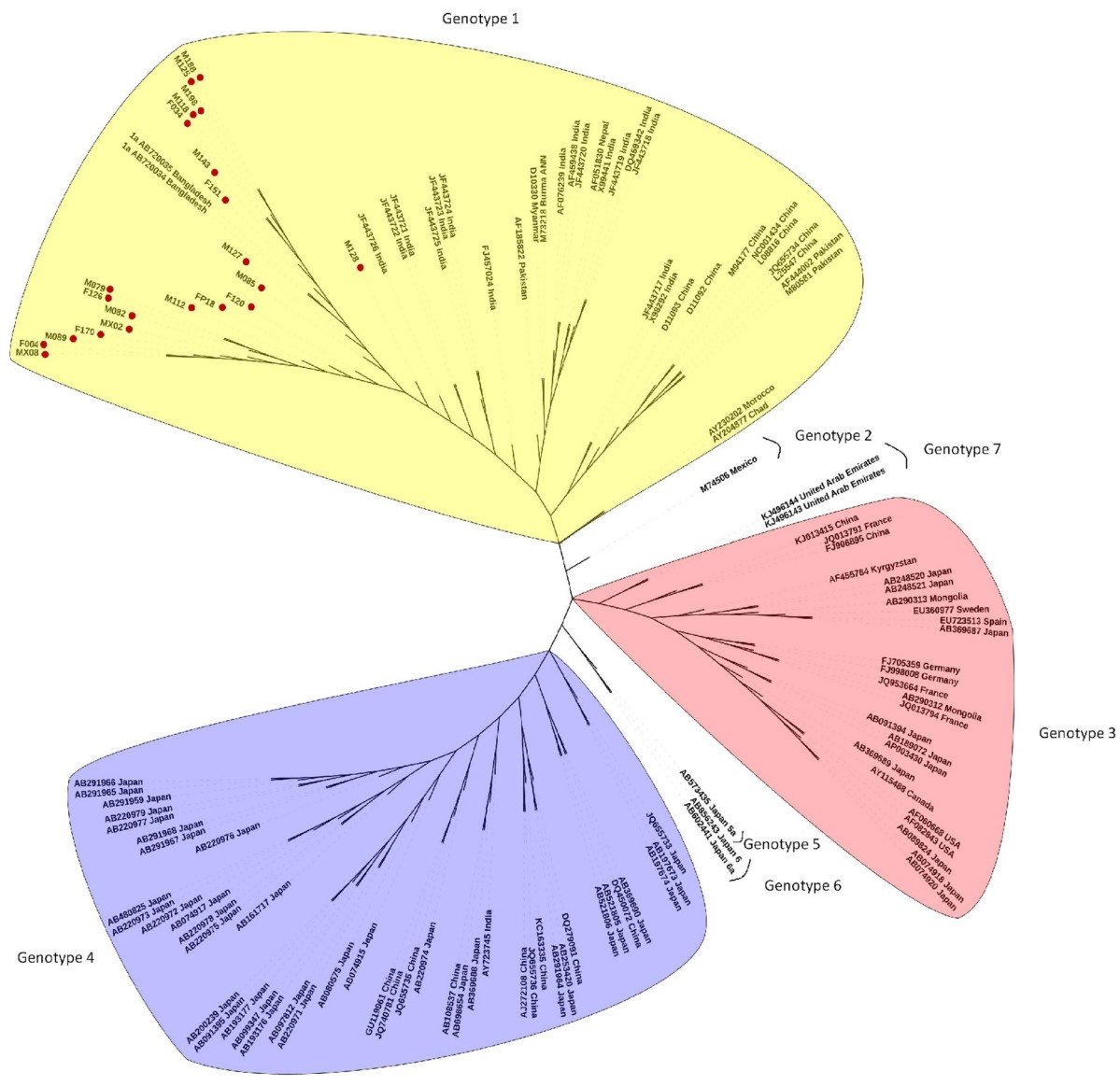

**Fig 1. Phylogenetic analysis of 109 reference HEV genome sequence and 21 Bangladeshi endemic HEV genome sequences from present syudy.** A midpoint rooted tree showing the relationship between the 21 Bangladeshi HEV genome sequences with 109 reference sequences representing all genotype and subtypes. The tree was constructed using RAxML v7.2.8 available in Geneious software using GTR+G+I nucleotide substitution model with 500 bootstrapping replicates. The Bangladeshi HEV strains from this study is presented as followed by strain number, and the reference genomes are presented as genotype, subtype followed by GenBank accession number and the country of origin. The scale bar indicates the number of nucleotide substitution.

D1384G, K1398N, and Y1587F mutations. With respect to immune escape mutations, all sequences had the L477T mutation associated with the B and T cell immune reactivity (Table 2). Additionally, all viruses had mutations associated with altered T cell antigenicity (MCHI and MCHII) and B cell epitopes, exceptions were the T3 and B3 epitopes where 76.1% (16/21) of the HEV had mutations in these loci (Table 2). Mutations in T cell recognition epitope (T3, T5, and T7) with the highest reduction antigenicity were observed in 76.1%, 100%, and 100% of the HEV sequences, respectively (Table 2).

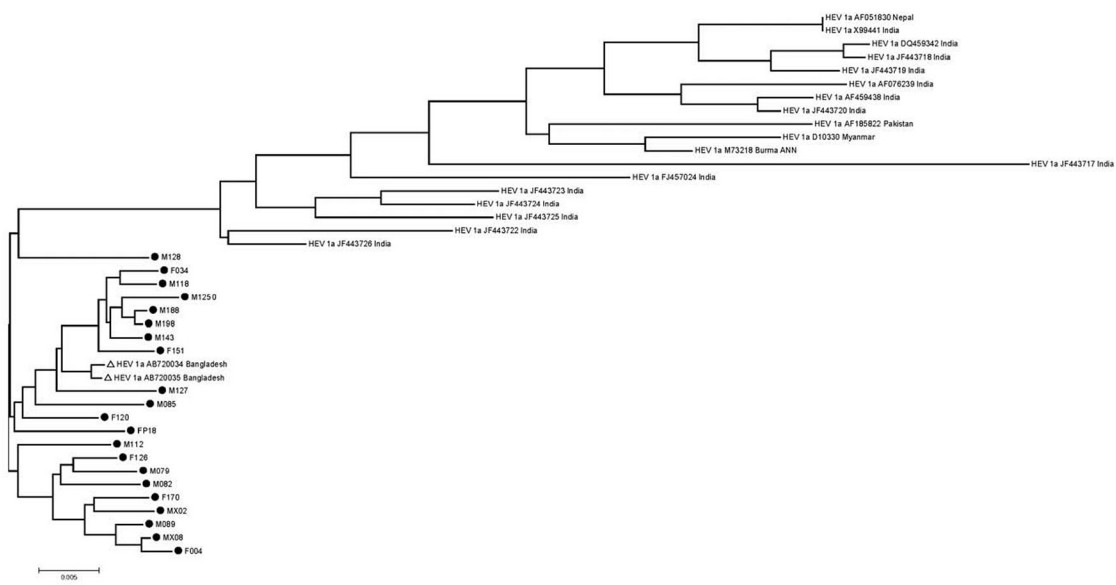

**Fig 2. Phylogenetic analysis of 20 reference HEV genome sequence from Bangladesh and neiboring countries and 21 Bangladeshi endemic HEV genome sequences from present syudy.** A midpoint rooted tree showing the relationship between the 21 Bangladeshi HEV genome sequences with 20 genotype 1a reference sequences from neighboring countries. The tree was constructed using RAxML v7.2.8 available in Geneious software using GTR+G+I nucleotide substitution model with 500 bootstrapping replicates. The Bangladeshi HEV strains from this study are presented as followed by strain number, and reference genomes are presented as genotype, subtype followed by GenBank accession number and the country of origin. The scale bar indicates the number of nucleotide substitution.

## Discussion

HEV outbreaks are commonly undetected in LMICs due to a lack of disease surveillance and/ or diagnostic capacity. Here, we analyzed samples collected systematically over a two year period to characterize endemic HEV genotypes in Bangladesh and compare with outbreak HEV genotypes. To our knowledge, this is the first report regarding the molecular characterization and WGS of endemic HEV strains from Bangladesh. Although there was no outbreak reported during the study period, the possibility of unidentified and/or unreported outbreak cannot be excluded. Smaller outbreaks often remain unreported due to lack of active surveillance in Bangladesh [30]; therefore, it is possible that viral strains characterized in this study may include outbreak strains.

Among the HEV IgM positive patients, 70% were HEV RNA positive. This discrepancy may be due to several factor including i) low viral load, ii) poor storage condition of the sample, or iii) the sensitivity of the qPCR assays for HEV isolates from Bangladesh. The prevalence of RNA positivity was higher in male patients compared to female (however, the difference was not significant) and was lowest in pregnant women. It is possible that male patients visited the hospital during active infection. In Bangladesh, men are main earning member in the family, and they seek healthcare earlier to minimize the wage loss due to sickness. Pregnant women in Bangladesh usually seek health care from "Maternal and Child Health (MCH)" clinics and referred to tertiary facilities when conditions worsen. This delay may result in lower RNA positivity in pregnant women.

All viral sequence in the present study were genotype 1a. When compared with genotype 1a strains from India, Nepal, Pakistan, and Myanmar, the Bangladeshi HEV strains formed a separate cluster within the genotype 1a cluster. HEV strains from the present study (and from the 2010 outbreak from northern Bangladesh) had high amino acid and DNA sequence similarity.

**Table 1. Mutations associated with FHF, chronic hepatitis, ribavirin treatment failure and B and T cell epitope neutralization in 21 HEV WGS isolated in this study.**

| Isolate No | Fulminant hepatic failure | | | | | | | | | Chronic hepatitis | Ribavirin treatment failure | | | | | | | Epitope neutralization | |
|---|---|---|---|---|---|---|---|---|---|---|---|---|---|---|---|---|---|---|---|
| | MT | Y-Dom | HVR | ORF1 | | | | | ORF2 | ORF1 | ORF1 | | | | | | ORF2 | ORF2 | |
| | | | | Helicase | | RDRP | | | | Helicase | | | RDRP | | | | | | |
| | F179S | A317T | T735I | L1110F | V1120I | F1439Y | C1483W | N1530T | P259S | V1213A | Y1320H | K1383N | D1384G | K1398N | V1479I | Y1587F | G1634K | L477T | L613T |
| | %(n) | %(n) | %(n) | %(n) | %(n) | %(n) | %(n) | %(n) | %(n) | %(n) | %(n) | %(n) | %(n) | %(n) | %(n) | %(n) | %(n) | %(n) | %(n) |
| | 0 (0) | 100 (21) | 100 (21) | 81.8 (17) | 100 (21) | 0 (0) | 0 (0) | 0 (0) | 100 (21) | 0 (0) | 0 (0) | 0 (0) | 0 (0) | 0 (0) | 100 (21) | 0 (0) | 100 (21) | 100 (21) | 0 (0) |
| BDHEV_F004 | | T | I | | I | | | | S | NA | | | | | I | | K | T | |
| BDHEV_F034 | | T | I | F | I | | | | S | NA | | | | | I | | K | T | |
| BDHEV_F 120 | | T | I | F | I | | | | S | NA | | | | | I | | K | T | |
| BDHEV_F 126 | | T | I | F | I | | | | S | NA | | | | | I | | K | T | |
| BDHEV_F 151 | | T | I | F | I | | | | S | NA | | | | | I | | K | T | |
| BDHEV_F170 | | T | I | F | I | | | | S | NA | | | | | I | | K | T | |
| BDHEV_P018 | | T | I | F | I | | | | S | NA | | | | | I | | K | T | |
| BDHEV_M 079 | | T | I | F | I | | | | S | NA | | | | | I | | K | T | |
| BDHEV_M 082 | | T | I | F | I | | | | S | NA | | | | | I | | K | T | |
| BDHEV_M 085 | | T | I | F | I | | | | S | NA | | | | | I | | K | T | |
| BDHEV_M 089 | | T | I | | I | | | | S | NA | | | | | I | | K | T | |
| BDHEV_M 112 | | T | I | F | I | | | | S | NA | | | | | I | | K | T | |
| BDHEV_M 118 | | T | I | F | I | | | | S | NA | | | | | I | | K | T | |
| BDHEV_M 125 | | T | I | F | I | | | | S | NA | | | | | I | | K | T | |
| BDHEV_M 127 | | T | I | F | I | | | | S | NA | | | | | I | | K | T | |
| BDHEV_M 128 | | T | I | F | I | | | | S | NA | | | | | I | | K | T | |
| BDHEV_M 143 | | T | I | F | I | | | | S | NA | | | | | I | | K | T | |
| BDHEV_M 188 | | T | I | F | I | | | | S | NA | | | | | I | | K | T | |
| BDHEV_M 198 | | T | I | F | I | | | | S | NA | | | | | I | | K | T | |
| BDHEV_MX02 | | T | I | F | I | | | | S | NA | | | | | I | | K | T | |
| BDHEV_MX08 | | T | I | | I | | | | S | NA | | | | | I | | K | T | |

MT: Methyl transferees; Y-Dom: Y domain; HVR: Hyper variable region; RDRP: RNA dependent RNA polymerase; NA: Not applicable.

**Table 2. Antigenicity evaluation of wild-type and mutated epitopes of 21 HEV strains from this study.**

| No of isolate % (N) | Region | Epitope number | Position | Predicted epitope aa | Antigenicity | Actual epitopes aa | Antigenicity |
|---|---|---|---|---|---|---|---|
| | | | | **MHC I** | | | |
| 100 (21) | ORF1 | T2 | T735I | ATPTPAAPL | 0.2 | ATP**T**/IPAAPL | 0.2 |
| 76.1 (16) | ORF1 | T3 | L1110F | TTSRVLRSL | 0.4 | TTSRV**L**/FRSL | -0.6 |
| 100 (21) | ORF1 | T5 | G1634K | AVSDFLRGL | 0.4 | AVSDFLR**G**/**K**L | -0.6 |
| | | | | **MHC II** | | | |
| 100 (21) | ORF1 | T7 | A317T | FHAVPAHIW | 0.5 | FHAVPA**A**/**T**HIW | -0.7 |
| 100 (21) | | T8 | T735I | IPSRAATPT | 0.2 | IPSRAATP**T**/**I** | 1 |
| 100 (21) | | T9 | V1120I | FWGEPAVGQ | 0.2 | FWGEPAV**V**/**I**GQ | 0.2 |
| 100 (21) | | T10 | V1479I | LGLECAVME | 0.8 | LGLECAV**V**/**I**ME | 0.7 |
| 100 (21) | | T11 | G1634K | LRGLTNVAQ | 0.7 | LR**G**/**K**LTNVAQ | 0.7 |
| 100 (21) | ORF2 | T12 | L477T | WLSLLAAEY | 1.1 | WLSL**L**/**T**AAEY | 1.7 |
| | | | | **B cell epitopes** | | | |
| 100 (21) | ORF1 | B2 | T735I | TPAAPLPSP | 0.1 | **T**/**I**PAAPLPSP | 0.1 |
| 76.1 (16) | | B3 | L1110F | TTSRVLRSL | 0.1 | TTSRV**L**/**F**RSL | 0.01 |
| 100 (21) | | B4 | V1120I | WGEPAVGQK | 0.8 | WGEPAV**V**/**I**GQK | 1 |
| 100 (21) | | B6 | G1634K | SDFLRKLTN | 0.1 | SDFLR**G**/**K**LTN | -0.1 |
| 100 (21) | ORF2 | B7 | L477T | SLTAAEYDQ | 1.1 | SL**L**/**T**AAEYDQ | 1.0 |
| 100 (21) | | B8 | L613T | LLDYPARAH | 0.4 | **L**/**T**LDYPARAH | 0.4 |

The effect of mutation and antigenicity was adopted from Ikram et al,. Bolt letters indicate mutations in the predicted epitopes.

We suspect that these HEV strains have entered endemic circulation and associated with sustained infections and HEV outbreaks. The exact etiology of outbreak or epidemic is beyond our investigation, but could be associated with various host or environmental factors. Most of the HEV sequences in our study harbored FHF and AVH associated mutations. It is possible that the majority of patients had severe infections as severely ill patients are generally referred to tertiary hospitals. These mutations have also been reported in HEV strains from FHF patients in India [16]. The clustering of Bangladeshi HEV strains among genotype 1a might be due possible evolution and host genetic adaptation of these viruses in this population [5].

The predicted B and T cells epitopes overlap in HEV, and these mutations might alter (increase or decrease) the antigenicity of the HEV. Mutations with highest antigenicity alteration in MHC class I (T3 and T5) and MCH II (T7) were present in most of the HEV in our study. It is possible that this is a strategy adopted by HEV to escape the host immune response. For MHC I, two immune reactivity related mutations (L477T and L613T) overlap with B and T cell epitopes. Generally, antibody recognizing conformational epitopes are neutralizing, and the amino acid residues L477 and L613 in the capsid protein might play a role in epitope deformation [31]. HEV specific T cell targets are generally conserved and located in the capsid protein in ORF2. As T cell response persists for a longer period, these mutations can influence both clearance of virus after primary infection and protective immune response against secondary infection [32]. The presence of G1634K mutation affects the ribavirin treatment response and a V1479I mutation is associated with ribavirin treatment failure. All HEV in our

study had a G1634K and a V1479I mutation. Therefore, it is possible that HEV patients in Bangladesh treated with ribavirin might respond poorly [17,18,33].

This study has several limitations; the samples were collected from patients attending a single tertiary care facility in Dhaka and may not be representative of the general population of Bangladesh. Patients with mild or moderate infection usually seek care at the local hospitals or private practitioners. As the samples were unlinked, clinical condition of the patients and disease outcome is unknown. We collected only 4 samples per months, increasing the sample size might have captured more variations present in the HEV strains in Bangladesh. Although there was no reported outbreak during the study period, the possibility of unreported outbreaks cannot be excluded.

## Conclusions

Taken together, our data show that strains of HEV genotype 1a are dominant in Bangladesh and are associated with endemic and outbreak of HEV infection. HEV isolates in Bangladesh have high prevalence of virulence associated and ribavirin resistant mutations which may influence treatment with Ribavirin. HEV isolates in our study contains mutation which alters antigenicity to B and T cell epitopes and facilitates evading host immune response.

## Supporting information

**S1 File. List of primers used for whole genome amplification in this study.**
(DOCX)

**S2 File. Accession number, genotype and subtype, and country of origin for 109 Hepatitis E virus used as representative reference strain.**
(DOCX)

**S3 File. Sociodemographic characteristics, HEV RNA positivity and HEV WGS information 92 patient samples analyzed in this study.**
(DOCX)

**S4 File. Schematic presentation of the sequencing of 92 HEV IgM ELISA positive samples.**
Among 92 samples, 70 were qPCR positive (column 3), qPCR negative samples are presented in transparent rows. A total of 825 fragments (column 28) could be amplified and sequenced (dark block) (range 1–24). Whole genome sequence could be obtained for 21 HEV viral strain. Fragment 20 (nucleotide sequence 5822 to 6211) contains the ORF 2–3 overlapping section could be sequence in 38 (21+ 17) strain.
(DOCX)

**S5 File. Phylogenetic analysis of 109 reference HEV genome sequence and 279bp fragment (ORF 2 and ORF) of 38 Bangladeshi endemic HEV sequence from present study.** A midpoint rooted tree showing the relationship between the 38 Bangladeshi HEV sequences with 109 reference sequences representing all genotype and subtypes. The tree was constructed using RAxML v7.2.8 available in Geneious software using GTR+G+I nucleotide substitution model with 500 bootstrapping replicates. The Bangladeshi HEV strains from this study is presented as ● followed by strain number, and the reference genomes are presented as genotype, subtype followed by GenBank accession number and the country of origin. The scale bar indicates the number of nucleotide substitution.
(DOCX)

## Acknowledgments

We thank OUCRU laboratory management staff for supporting the sequencing of HEV strains.

## Author Contributions

**Conceptualization:** Saif Ullah Munshi, Tahmina Akther, Shahina Tabassum, Nilufa Parvin, Stephen Baker, Motiur Rahman.

**Data curation:** Trang Nguyen Hoa, Chau Le Ngoc, Thanh Tran Thi Thanh.

**Formal analysis:** Trang Nguyen Hoa, Khanh Nguyen Ngoc, Thanh Tran Thi Thanh.

**Funding acquisition:** Motiur Rahman.

**Investigation:** Trang Nguyen Hoa, Saif Ullah Munshi, Khanh Nguyen Ngoc, Chau Le Ngoc, Nilufa Parvin.

**Methodology:** Trang Nguyen Hoa, Saif Ullah Munshi, Khanh Nguyen Ngoc, Chau Le Ngoc, Thanh Tran Thi Thanh.

**Project administration:** Trang Nguyen Hoa, Saif Ullah Munshi, Tahmina Akther, Shahina Tabassum, Motiur Rahman.

**Resources:** Thanh Tran Thi Thanh, Shahina Tabassum, Nilufa Parvin, Stephen Baker, Motiur Rahman.

**Supervision:** Thanh Tran Thi Thanh, Tahmina Akther, Nilufa Parvin, Stephen Baker.

**Validation:** Khanh Nguyen Ngoc, Chau Le Ngoc, Shahina Tabassum.

**Writing – original draft:** Stephen Baker, Motiur Rahman.

**Writing – review & editing:** Trang Nguyen Hoa, Saif Ullah Munshi, Khanh Nguyen Ngoc, Chau Le Ngoc, Thanh Tran Thi Thanh, Tahmina Akther, Shahina Tabassum, Nilufa Parvin, Stephen Baker, Motiur Rahman.

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
