## [Decision Letter · Decision Letter 0]

24 Feb 2021

PONE-D-20-39475

A tightly clustered hepatitis E virus genotype 1a is associated with endemic and outbreak infections in Bangladesh

PLOS ONE

Dear Dr. Rahman,

Thank you for submitting your manuscript to PLOS ONE. After careful consideration, we feel that it has merit but does not fully meet PLOS ONE’s publication criteria as it currently stands. Therefore, we invite you to submit a revised version of the manuscript that addresses the points raised during the review process.

Please provide the details requested by the reviewer and explain the process of the full sequence validation using RACE PCR. As noted the abstract line 40 is confusing as we may understand that 38+21 strains were typed while there were 38 genotyped strains from which 21 full sequences were obtained.

We look forward to receiving your revised manuscript.

Kind regards,

Pierre Roques, Ph.D.

Academic Editor

PLOS ONE

Journal Requirements:

2. As part of your revisions please provide additional information about the source/origin of the samples utilized for this study. You indicate in your Methods that you collected samples that had been submitted to the virology department of BSMMU and that these samples had been systematically selected, i.e.  Systematically first four HEV IgM ELISA positive patients each month.

Please indicate whether the patients or next of kin provided informed consent for the use of these samples and the form of consent (written or oral). If the samples were anonymized please state so.

Thank you for your attention in this matter.

'This study was supported by the Oxford University Clinical Research Unit, Vietnam.'

'The author(s) received no specific funding for this work.'

Additional Editor Comments:

sorry for the delay in the reviewing process.

Reviewers' comments:

Reviewer's Responses to Questions

**Comments to the Author**

1. Is the manuscript technically sound, and do the data support the conclusions?

Reviewer #1: Yes

2. Has the statistical analysis been performed appropriately and rigorously? 

Reviewer #1: I Don't Know

3. Have the authors made all data underlying the findings in their manuscript fully available?

Reviewer #1: Yes

4. Is the manuscript presented in an intelligible fashion and written in standard English?

Reviewer #1: Yes

5. Review Comments to the Author

Reviewer #1: Hao and colleagues describe the closely grouped hepatitis E virus genotype 1a associated with endemic and outbreak infections in Bangladesh. The paper is quite interesting, I have some concerns.

Line 40 ; Lines 43-44 ; lines-239-240 : Authors should provide precisions at this level for a better understanding. Is it the 21 sequences from complete genome sequence or the 38 sequences from incomplete genome sequence.

Lines 121-124 : The authors should provide more details on the selection criteria for the first four ELISA-positive patients of the month, in order to better interpret the results, e.g., did these patients have any symptoms of the disease or not ?

Line 129 : Review the title of the section, it's « Amplification and sequencing »

Lines 134-137 : What was the point of doing this RACE PCR, given that only the 21 sequences underwent a phylogenetic analysis?

Lines 143-144 : What is the point to aligne overlapping sequences?

Lines 194-195 : The authors report that at least one DNA fragment could be sequenced in all plasma samples. However, only 38 sequences ((17 from incomplete genome sequence and 21 from complete genome sequence) on the 70 PCR positive samples could be obtained and sequenced.

Lines 197-198 : How do the authors explain this difference with the sequences obtained by WGS (21 complete genome sequences) and RACE PCR (38 incomplete genome sequences) ?

Lines 210-126 : Is this high sequence conservation observed only at the ORF1 sequences level ?

Lines 275-276 : Aren't the two arguments reported by the authors linked?

Lines 276-282 : These explanations of the authors are in contradiction with those reported at the level of the results (lines 189-190).

Lines 283-285 : How do the authors explain this distinction of the Bangladeshi HEV strains within the genotype 1a cluster ?

Lines 290-292 : Doesn't this explanation contradict the argument that the observed discrepancy between IgM and RNA-positive levels is due to a resolution of the active infection ?

Line 38 : Can it be said that strains of genotype 1a are dominant in Bangladesh, since the authors themselves reported that their study had several limitations

6. PLOS authors have the option to publish the peer review history of their article (what does this mean?). If published, this will include your full peer review and any attached files.

Reviewer #1: No

---

## [Author Response · Author response to Decision Letter 0]

31 Mar 2021

Response to reviewer’s:

PONE-D-20-39475

A tightly clustered hepatitis E virus genotype 1a is associated with endemic and outbreak infections in Bangladesh

PLOS ONE

Dear Dr. Rahman,

Thank you for submitting your manuscript to PLOS ONE. After careful consideration, we feel that it has merit but does not fully meet PLOS ONE’s publication criteria as it currently stands. Therefore, we invite you to submit a revised version of the manuscript that addresses the points raised during the review process.

Please provide the details requested by the reviewer and explain the process of the full sequence validation using RACE PCR. As noted the abstract line 40 is confusing as we may understand that 38+21 strains were typed while there were 38 genotyped strains from which 21 full sequences were obtained.

Response: We have rephrased the sentence and warding to address the confusion on the number of strains genotyped and the number of strains with whole genome sequenced. Plz see line 38-42.

Response: We have uploaded separate file as “Response to Reviewers”.

Response: We have uploaded a file labeled “Revised Manuscript with Track Change”.

Response: We have uploaded a file labeled “Manuscript”.

Response: We have included the updated financial disclosure statement in Cover letter. 

We look forward to receiving your revised manuscript.

Kind regards,

Pierre Roques, Ph.D.

Academic Editor

PLOS ONE

Journal Requirements:

 Response: We have followed the guidelines

2. As part of your revisions please provide additional information about the source/origin of the samples utilized for this study. You indicate in your Methods that you collected samples that had been submitted to the virology department of BSMMU and that these samples had been systematically selected, i.e. Systematically first four HEV IgM ELISA positive patients each month.

Please indicate whether the patients or next of kin provided informed consent for the use of these samples and the form of consent (written or oral). If the samples were anonymized please state so.

Response: We have revised the section as suggested by the reviewer and mentioned the how samples were anonymized. Plz see line 124-129.

Thank you for your attention in this matter.

'This study was supported by the Oxford University Clinical Research Unit, Vietnam.'

'The author(s) received no specific funding for this work.'

Response: We have removed the funding information from acknowledgement section. We have added the amended information in the revised cover letter. We have added new text in acknowledgement section.

Additional Editor Comments:

sorry for the delay in the reviewing process.

Response: We have reviewed the references and couldn’t find any retracted article. 

Reviewers' comments:

Reviewer's Responses to Questions

Comments to the Author

1. Is the manuscript technically sound, and do the data support the conclusions?

Reviewer #1: Yes

2. Has the statistical analysis been performed appropriately and rigorously? 

Reviewer #1: I Don't Know

3. Have the authors made all data underlying the findings in their manuscript fully available?

Reviewer #1: Yes

4. Is the manuscript presented in an intelligible fashion and written in standard English?

Reviewer #1: Yes

5. Review Comments to the Author

Reviewer #1: Hao and colleagues describe the closely grouped hepatitis E virus genotype 1a associated with endemic and outbreak infections in Bangladesh. The paper is quite interesting, I have some concerns.

Line 40 ; Lines 43-44 ; lines-239-240 : Authors should provide precisions at this level for a better understanding. Is it the 21 sequences from complete genome sequence or the 38 sequences from incomplete genome sequence.

Response: We have rephrased the sentence as following; “A 279 bp open reading frame (ORF) 2 and ORF 3 sequence was obtained from 54.2% (38/70) of the strains. Of these, whole genome sequence (WGS) was obtained from 21 strains. In phylogenetic analysis using the 279 bp sequence all HEV sequences belonged to genotype 1 and subtype 1a” . Plz see Abstract line 38 to 42. 

Lines 121-124 : The authors should provide more details on the selection criteria for the first four ELISA-positive patients of the month, in order to better interpret the results, e.g., did these patients have any symptoms of the disease or not ?

Response: We agree with the reviewer that clinical information of the patients would have been helpful to interpret the data. However, this component of the study was a genome surveillance for HEV to understand the genomic epidemiology and genotype of HEV strains circulating in Bangladesh. Patients with HEV IgM test request and HEV IgM positive were eligible for including in the study. We selected the first four HEV IgM positive samples each month. Samples were unlinked with patient data be replacing the patient ID with a study ID. We had limited patient information e.g. age, gender and pregnancy status (in case of female) of the patients. This has been mentioned in the limitations of the study. We have revised study design and population section. Plz see line 122 to 129 and line 343.

Line 129 : Review the title of the section, it's « Amplification and sequencing »

Response: We have revised the title of the section.

Lines 134-137 : What was the point of doing this RACE PCR, given that only the 21 sequences underwent a phylogenetic analysis? 

Response: We apologies for the confusion. RACE PCR was not performed for all strains. In strains where maximum number of overlapping fragments could be amplified and sequenced a RACE PCR was conducted to complete the 5’ and 3’ region. Number of fragments amplified and sequenced for each strain are presented “S File 4; fragment map”. We have also revised the text; Plz see line 140-142 

Lines 143-144 : What is the point to aligne overlapping sequences? 

Response: For each strain we tried to amplify and sequence the whole genome in 24 fragments. Overlapping sequences were aligned to generate the full length sequences where all fragments could be amplified and sequenced (n=21). The number of fragments amplified and sequenced are presented in “S File 4; fragment map”. We have revised the warding to clarify it. Plz see line 149-150 

Lines 194-195 : The authors report that at least one DNA fragment could be sequenced in all plasma samples. However, only 38 sequences ((17 from incomplete genome sequence and 21 from complete genome sequence) on the 70 PCR positive samples could be obtained and sequenced. 

Response: We apologies for the confusion. We attempted to sequence complete genome of all strains (n=70), however we could not amplify and sequence all 24 fragments for each stains. Fragments amplified and sequenced for each strain is presented “S file 4”. We could amplify 825 fragments from 70 strains. Of these, all 24 fragments were amplified and sequenced for 21 strains. All 24 fragments could not be amplified or sequenced for the remaining 49 strains. Of these strains (n=49), along with other fragments, the 279 bp ORF2 and ORF 3 region (Fragment no 20 (S20)) could be amplified and sequenced for 17 strains. 

Lines 197-198 : How do the authors explain this difference with the sequences obtained by WGS (21 complete genome sequences) and RACE PCR (38 incomplete genome sequences) ? 

Response: RACE PCR was not conducted for strains with incomplete genome sequences. We conducted RACE PCR for 21 stains (21 complete genome sequence) to complete the 5’ and 3’ ends. We have revised the text on this, plz see line 140-142 

Lines 210-126 : Is this high sequence conservation observed only at the ORF1 sequences level ?

Response: We observed high sequence conservation in all three region among isolates where WGS were available. 

Lines 275-276 : Aren't the two arguments reported by the authors linked?

Response: We have revised the arguments. Plz see line 285. 

Lines 276-282 : These explanations of the authors are in contradiction with those reported at the level of the results (lines 189-190). 

Response: We apologies for the confusion. In Line 189-190, we compared all patients (both male and female) recruited during 2013-14 and 2014-15. In line 276-282, we compared difference among male and female patients. We have rephrased the sentence and added data on RNA positivity among male and female patients in results section. Plz see line 194-200. 

Lines 283-285 : How do the authors explain this distinction of the Bangladeshi HEV strains within the genotype 1a cluster ? 

Response: The exact reason for this clustering among HEV genotype 1a strains is unknown, but it is possible that viral evolution and host genetic adaption of these viruses in Bangladeshi population might be an underlying reason. We have included a comment in the discussion, plz see line 302-303 

Lines 290-292 : Doesn't this explanation contradict the argument that the observed discrepancy between IgM and RNA-positive levels is due to a resolution of the active infection ?

Response: We agree with the reviewer and removed the sentence: i) active HEV infection was resolved when the patient visited the hospital. Plz see line 285.

Line 318 : Can it be said that strains of genotype 1a are dominant in Bangladesh, since the authors themselves reported that their study had several limitations

Response: we have made the suggestive change. Plz see line 328

6. PLOS authors have the option to publish the peer review history of their article (what does this mean?). If published, this will include your full peer review and any attached files.

Do you want your identity to be public for this peer review? For information about this choice, including consent withdrawal, please see our Privacy Policy.

Reviewer #1: No

Response: We have uploaded and the figures in PACE and the figures meet PLOS requirements

---

## [Decision Letter · Decision Letter 1]

30 Apr 2021

PONE-D-20-39475R1

A tightly clustered hepatitis E virus genotype 1a is associated with endemic and outbreak infections in Bangladesh

PLOS ONE

Dear Dr. Rahman,

Thank you for submitting your manuscript to PLOS ONE. After careful consideration, we feel that it has merit but does not fully meet PLOS ONE’s publication criteria as it currently stands. Therefore, we invite you to submit a revised version of the manuscript that addresses the points raised during the review process.

Please provide to the reviewers and include in the article text the answers to his questions.

We look forward to receiving your revised manuscript.

Kind regards,

Pierre Roques, Ph.D.

Academic Editor

PLOS ONE

Journal Requirements:

Reviewers' comments:

Reviewer's Responses to Questions

**Comments to the Author**

1. If the authors have adequately addressed your comments raised in a previous round of review and you feel that this manuscript is now acceptable for publication, you may indicate that here to bypass the “Comments to the Author” section, enter your conflict of interest statement in the “Confidential to Editor” section, and submit your "Accept" recommendation.

Reviewer #1: (No Response)

2. Is the manuscript technically sound, and do the data support the conclusions?

Reviewer #1: Yes

3. Has the statistical analysis been performed appropriately and rigorously? 

Reviewer #1: I Don't Know

4. Have the authors made all data underlying the findings in their manuscript fully available?

Reviewer #1: Yes

5. Is the manuscript presented in an intelligible fashion and written in standard English?

Reviewer #1: Yes

6. Review Comments to the Author

Reviewer #1: Some of my concerns about the study still remain.

Line 31: Authors should modify "HEV infected patients" to HEV immunoglobulin M (IgM) positive patients

Lines 40-42: The authors should clarify, because it seemed to me that the phylogenetic analysis was performed with the 21 WGS sequences of HEV (Figure 1 and 2)

Line 132: In this section, what was the methodology used to obtain the 38 sequences of 279 bp.

Line 137: To Clarify the nested PCR method used: What primers were used? What were the expected fragment sizes?

Lines 167-168: The authors must specify that it is on the 21 HEV WGS that the mutations have been identified.

Lines 225-227: these data are not available in the figures (Fig 1 and 2).

Lines 264-267: The authors should note that the analysis of virulence-associated mutations was limited to the 21 WGS (Table 1)

Lines 300-302: Molecular detection tests (qPCR) by reference are more sensitive than serological tests (Ig anti-HEV). Thus, the difference observed in this study is not rather related to the non-existence of a serological test of reference.

Lines302-307: The authors must specify that this observed difference was not significant (P=0.08)

7. PLOS authors have the option to publish the peer review history of their article (what does this mean?). If published, this will include your full peer review and any attached files.

Reviewer #1: No

---

## [Author Response · Author response to Decision Letter 1]

9 May 2021

Review Comments to the Author and response

Reviewer #1: Some of my concerns about the study still remain.

Line 31: Authors should modify "HEV infected patients" to HEV immunoglobulin M (IgM) positive patients

Response: We have modified the sentence as suggested, plz see line 31

Lines 40-42: The authors should clarify, because it seemed to me that the phylogenetic analysis was performed with the 21 WGS sequences of HEV (Figure 1 and 2)

Response: We have modified the sentence to clarify it, plz see line 41-44

Line 132: In this section, what was the methodology used to obtain the 38 sequences of 279 bp.

Response: We apologize for the confusion. The 279 bp fragment was amplified and sequenced by primer set 20 (S1 File). We have added clarification, plz see line 145-147 

Line 137: To Clarify the nested PCR method used: What primers were used? What were the expected fragment sizes?

Response: For nested PCR we used 20 sets of primers (each set contains 4 primers (two external and two internal). Plz see details in S 1 File. The fragment size was 450-600 bp, plz see line 139-140. 

Lines 167-168: The authors must specify that it is on the 21 HEV WGS that the mutations have been identified.

Response: We have specified as suggested. Plz see line 172-173

Lines 225-227: these data are not available in the figures (Fig 1 and 2).

Response: We agree with the reviewer. We have added a supplementary figure (S4 File) to present the data. 

Lines 264-267: The authors should note that the analysis of virulence-associated mutations was limited to the 21 WGS (Table 1)

Response: We have mentioned in the title of Table 1. Plz see table 1 and table 2

Lines 300-302: Molecular detection tests (qPCR) by reference are more sensitive than serological tests (Ig anti-HEV). Thus, the difference observed in this study is not rather related to the non-existence of a serological test of reference.

Response: We have modified the sentence to address the issue. Plz see line 308 

Lines302-307: The authors must specify that this observed difference was not significant (P=0.08)

Response: We have revised the sentence. Plz see line 309

---

## [Decision Letter · Decision Letter 2]

25 May 2021

PONE-D-20-39475R2

A tightly clustered hepatitis E virus genotype 1a is associated with endemic and outbreak infections in Bangladesh

PLOS ONE

Dear Dr. Rahman,

Thank you for submitting your manuscript to PLOS ONE. After careful consideration, we feel that it has merit but does not fully meet PLOS ONE’s publication criteria as it currently stands. Therefore, we invite you to submit a revised version of the manuscript that addresses the points raised during the review process.

Please correct the minor points rised by the reviewer and correct the supplement table 1 with the primers as the indication of sense and antisens are misleaded and confusing. In addition, numbering of the reverse-sens primers should be also reversed as an example

HEV-S2-A1

1st antisense

**GTTCAGCGYTGGTATACTGC**

348 - 367

should be numbered as 367 - 348 to  indicate that the sequence is read on the negative (-) strand ; in extenso in the 3' to 5' direction on the positive (+) strand that is the reference sequence used to number the position.

In this S1 file all primers' sets are for nested PCR and named it 1st sense, 2nd sense then 1st antisense, 2nd antisense but the numbering you indicated clearly shown that the order is 1st sense, 1st antisense, 2nd sense, 2nd antisense. Please clarify and explain the size of the Set 20 product (ie how you read 279 base pairs at the end).

We look forward to receiving your revised manuscript.

Kind regards,

Pierre Roques, Ph.D.

Academic Editor

PLOS ONE

Journal Requirements:

Reviewers' comments:

Reviewer's Responses to Questions

**Comments to the Author**

1. If the authors have adequately addressed your comments raised in a previous round of review and you feel that this manuscript is now acceptable for publication, you may indicate that here to bypass the “Comments to the Author” section, enter your conflict of interest statement in the “Confidential to Editor” section, and submit your "Accept" recommendation.

Reviewer #1: (No Response)

2. Is the manuscript technically sound, and do the data support the conclusions?

Reviewer #1: Yes

3. Has the statistical analysis been performed appropriately and rigorously? 

Reviewer #1: I Don't Know

4. Have the authors made all data underlying the findings in their manuscript fully available?

Reviewer #1: Yes

5. Is the manuscript presented in an intelligible fashion and written in standard English?

Reviewer #1: Yes

6. Review Comments to the Author

Reviewer #1: Some of my concerns about the study still remain.

Lines 60-61: Authors should modify "HEV infected patients" to HEV immunoglobulin M (IgM) positive patients

Lines 146-147: How do the authors explain the obtaining of a 279bp fragment, when the primers used amplify fragments of 450-550

Line 146: to be corrected ORF3

Lines 305-307: Molecular detection tests (qPCR) by reference are more sensitive than serological tests (Ig anti-HEV). Thus, the difference observed in this study is not rather related to the non-existence of a serological test of reference.

Lines 316-317: Despite this high sequence similarity, the authors report in the summary section that the HEV strains from the present study formed a distinct cluster with the 2010 HEV outbreak strains from northern Bangladesh lines (45-46)

7. PLOS authors have the option to publish the peer review history of their article (what does this mean?). If published, this will include your full peer review and any attached files.

Reviewer #1: No

---

## [Author Response · Author response to Decision Letter 2]

7 Jul 2021

Response to Reviewer:

Please correct the minor points raised by the reviewer and correct the supplement table 1 with the primers as the indication of sense and antisense are misleading and confusing. 

In addition, numbering of the reverse-sense primers should be also reversed as an example

HEV-S2-A1 1st antisense GTTCAGCGYTGGTATACTGC 348 - 367

should be numbered as 367 - 348 to indicate that the sequence is read on the negative (-) strand ; in extenso in the 3' to 5' direction on the positive (+) strand that is the reference sequence used to number the position.

In this S1 file all primers' sets are for nested PCR and named it 1st sense, 2nd sense then 1st antisense, 2nd antisense but the numbering you indicated clearly shown that the order is 1st sense, 1st antisense, 2nd sense, 2nd antisense. 

Response 

We have revised the S1 File and corrected the antisense primer position numbering. The primer sequence is already reverse. We have included the the reference sequence used to number the positions. 

We have corrected the organization of primers as 1st sense, 1st antisense, 2nd sense, 2nd antisense. 

Please clarify and explain the size of the Set 20 product (i.e how you read 279 base pairs at the end).

Response:

The fragment 20 will amplify a `440bp fragment. Of these 440 bp fragment, we extracted 279 bp for genotyping (Ref 26). We have revised the sentence as follows and added a new reference. “We have revised the sentence for clarification “The entire genome sequences (amplified by primer set 1 to 20 (S1 File)) and a 279bp region (position 5972-6319 of GenBank accession no M73218) of ORF2 and ORF3 (part of the `440 bp fragment amplified by primer set 20 (S1 File)) were used for final analysis [26]” plz see page 145-148.

---

## [Editor Report · Decision Letter 3]

9 Jul 2021

A tightly clustered hepatitis E virus genotype 1a is associated with endemic and outbreak infections in Bangladesh

PONE-D-20-39475R3

Dear Dr. Rahman,

We’re pleased to inform you that your manuscript has been judged scientifically suitable for publication and will be formally accepted for publication once it meets all outstanding technical requirements.

Kind regards,

Pierre Roques, Ph.D.

Academic Editor

PLOS ONE
---

## [Editor Report · Acceptance letter]

13 Jul 2021

PONE-D-20-39475R3 

A tightly clustered hepatitis E virus genotype 1a is associated with endemic and outbreak infections in Bangladesh 

Dear Dr. Rahman:

I'm pleased to inform you that your manuscript has been deemed suitable for publication in PLOS ONE. Congratulations! Your manuscript is now with our production department. 

Kind regards, 

on behalf of

Dr. Pierre Roques 

Academic Editor

PLOS ONE